# Co-Production Within Academic Constraints: Insights from a Case Study

**DOI:** 10.3390/ijerph21111503

**Published:** 2024-11-13

**Authors:** Evelyn Callahan, Niamh Murtagh, Alison Pooley, Jenny Pannell, Alison Benzimra

**Affiliations:** 1The Bartlett School of Sustainable Construction, University College London (UCL), London WC1E 7HB, UK; e.callahan@ucl.ac.uk (E.C.); jennypannell@virginmedia.com (J.P.); 2Suffolk Sustainability Institute, University of Suffolk, Ipswich IP4 1QJ, UK; a.pooley@uos.ac.uk; 3United St Saviour’s Charity, London SE1 3JW, UK; abenzimra@ustsc.org.uk

**Keywords:** co-production, co-research, almshouse, housing, power, social

## Abstract

Co-production in research offers the potential for multiple benefits, including amplifying the voices of the marginalised, reducing power inequalities between academic researchers and co-researchers outside of academia, increased likelihood of impact, and improvement in the research process. But alongside increased interest in co-production, there is increased awareness of its contextual constraints. Key amongst these are institutional orthodoxies in academia, including time-limited, project-based research and precarious employment for junior researchers. To examine how the potential benefits of co-production can be achieved within the constraints of current academic systems, a case study project was assessed against a documented set of expectations for the co-production of research with older adults. The case study was a research project conducted with seven almshouse communities in England on the topic of social resilience. The wider almshouse communities—staff, trustees, and residents—were involved in co-production. The assessment concluded that co-production led to rich data and deep understanding. Co-production aided the development of skills and experiences of the co-researchers, resulted in changes in practice, and challenged power differentials, albeit in limited ways, but could not ensure the sustainability of relationships or impact. Key elements for effective co-production included the approach to and governance of the project, the formation of a Residents Advisory Group, and planning for the limited commitment that individuals and organisations outside of academia may be able to contribute to research.

## 1. Introduction

The push for co-production in social and health research has been underway for decades. In essence, co-production is a participatory approach to research [1] that involves working with (as opposed to on or for) non-academic participants. Co-production approaches aim to render research more democratic. In the United Kingdom (UK), the national funding co-ordinator for health and care research placed public and patient involvement in research as a central policy from its inception in 2006 [2]. Possibly led by initiatives in health research, a wider ‘turn to’ co-production across disciplines has been observed more recently [1,3,4]. In research on ageing, co-production has become increasingly common, helping to challenge the prevalent discourse of age as problematic and associated with ill health, and aligning with perspectives on active and successful ageing [5]. In addition, the aim of co-production to render research more democratic resonates with the recognition of older people as subject to prejudice and discrimination across many cultures [6]. Multiple benefits of co-production have been proposed but, equally, constraints have increasingly been recognised. Such constraints must be investigated and ways of mitigation sought so that the co-production of solutions becomes increasingly widespread to address the issues that are important to people as they age, across social, health, and economic domains, and particularly including housing and its social contexts. While there has been much theoretical discussion of co-production, as we will discuss below, there is relatively little literature focusing on the pragmatic issues confronting scholars who are faced with adopting this approach to research. The current paper aims to address this gap and to present practical insights on the use of co-production in research. In particular, academic constraints to effective co-production are considered and approaches for mitigation described, using an empirical case study for critical reflection on methodological decisions on co-production.

### 1.1. Benefits of Co-Production

Common aspirational benefits for co-production include sharing power between academic and non-academic researchers, valuing knowledge from everyone, involving the people on whom decisions will have impact, and making research respectful, equal, and of genuine benefit to citizens [7]. The emphasis for many scholars of the theory of co-production is on challenging existing societal power structures, and in particular, equalising power relations between academic and non-academic co-researchers [8,9]. It has been proposed that “those who have been most adversely affected by injustice must lead research collectives or be key decision makers” [10], p. 435. A key element has been the valuing of knowledge from outside of academia and recognising non-academic collaborators as ‘experts by experience’ [11]. Co-production can lead to richer findings, better achievement of research goals, and the maximisation of limited resources [7]. The potential benefits can be categorised as technocratic, that is, leading to higher-quality research and more impact, or democratic, that is, a normative approach with aims of shifting power relationships [12]. More specifically, the aims and advantages of co-production can be (i) substantive, to improve the quality of research; (ii) instrumental, so that the research will be used; (iii) normative, offering intrinsic value; or (iv) political, leading to empowerment [13]. Sharing power and valuing knowledge from all team members necessitates building and maintaining relationships [7], and benefits can emerge from research rather than be identified a priori [14]. For non-academic researchers, involvement in co-produced research can bring increased confidence, enhanced advocacy skills, social empowerment, and a sense of collective purpose, as well as the potential for sustained impact of the research [15]. Indeed co-production has been argued to be utopian in the sense that it can result in a transformed reality [3].

For research on ageing, the consensus that co-production should help to reduce power differentials is particularly apposite, given the recognition of older adults as a disempowered group [15]. Co-production can offer older adults more control of the process of research [16], and the benefits include a deeper understanding of issues for older people, building capacity for individuals and community through the development of skills and experiences, creating change in community, services, practices, or policies, and reducing marginalisation through empowerment [5]. Although the number of co-produced studies with older adults remains small as yet [5], there is growing recognition of the value such studies could bring to the field [17,18].

### 1.2. Meanings of Co-Production

Approaches to co-production vary in terms of meaning, scope, and aims. Although commonly espoused aims for co-production can be articulated, in fact, the term has been used as an umbrella term encompassing a wide range of approaches to research [3,5,13], and even described as a buzzword [19]. Related and often overlapping terms include patient and public involvement (PPI), co-research, participatory action research, and critical participatory action research. It can be suggested that co-research may be the loosest term, encapsulating most attempts to involve non-academic partners in research beyond the role of participant, and critical participatory action research as the most rigorous, with an epistemology grounded on the assumption of deep inequality [10]. One attempt at categorisation distinguishes between degrees of co-production, running from ‘thin’ to ‘full embeddedness’ [19]. A particularly insightful and comprehensive definition comes from the Co-Production Collective: “Co-production is a concept based on principles, a research method, a complex intervention, a challenge to academic conventions and systems, and a challenge to the self” [14], p. 55. In this paper, we have adopted the approach to co-production presented by James and Buffel [5], based on their systematic literature review of co-production in research with older adults. They noted four potential areas of benefit: (i) reaching deeper understanding from improved data; (ii) development of skills and experiences of co-researchers; (iii) creating change in the community, practices, services, or policies; and (iv) challenging power dynamics. In addition, they identified three areas of challenge: (i) power differentials; (ii) different expectations; and (iii) the sustainability of engagement and resources over time. Their contribution offers a succinct but comprehensive categorisation of the often long and unwieldly set of factors described elsewhere in the literature [14,15].

### 1.3. Critiques of Co-Production

Alongside the wide range of understandings of co-production are a number of critiques and reviews of barriers in achieving its goals. It has been argued that the initial enthusiasm for co-production as opening up new ways of knowledge production has evolved into a more nuanced understanding of limitations [20]. Co-production has been seen as simply a tick-box exercise in some cases [1,15], and it is thought that the very different requirements of working with non-academic organisations versus marginalised groups are often confounded [12]. A core criticism has been of attempts to examine co-production without recognition of its context. Co-production is always a product of its context [21,22]; for example, the funding context has been found to restrict what is possible in participative action research with older adults [23]. Paylor and McKevitt [1] argued that failures in the involvement of publics beyond academia are a result of context and pointed specifically to the logics of consumerism, managerialism, and performativity in which current research is embedded. While the consumerist notion of ‘customer’ is perhaps less relevant in research with older adults compared to more general health research, managerialist and associated performative logics prevail in academia, including in ageing research, with career progression dependent on individualised measurements of ‘outputs’, such as published papers and grant funding won [1,3].

A strong critique of co-production put forward by Oliver et al. [13] included the dearth of evidence for benefits alongside increasing acknowledgement of costs, such as additional administration and emotional labour, and the lack of recognition in terms of academic career progression. However, this argument was refuted by Williams and colleagues [12], who demonstrated that it focused on technocratic or instrumental objectives of research and failed to give equal consideration to the democratic rationale of co-production. In addition, a number of other scholars have pointed to institutional contexts as causal factors of ‘failed’ co-production [23,24,25].

Of these contexts, academic culture and norms have been particularly emphasised [14]. There is acknowledgement that, despite the democratic ideals of co-production, genuine sharing of power is not possible [8]: power differentials cannot be set aside for a co-produced research project but required for all other aspects of academic career performance [25]. Power hierarchies within academia emerge in part from the ubiquitous use of short-term contracts for the typically junior researchers most directly involved in co-production in research [25,26]. Precarity in the academy itself serves to exclude many from involvement in research, to the detriment of knowledge, learning, and society [27]. The ‘publish or perish’ demands on academics—part of the performative logic described by Paylor and McKevitt [1]—may be incompatible with the needs of co-production stakeholders. This can lead to tension between a research focus which reflects the priorities of all stakeholders and a sufficiently novel contribution to knowledge that will be publishable for the academic researchers [13]. The project-based, time-limited structure of funding gives no opportunity to reflect on learning and makes the development and maintenance of long-term relationships with co-production stakeholders almost impossible [1,13,25].

### 1.4. Aims and Objectives of This Paper

Recognising that the democratic and transformative ideals of co-production are valuable aspirations, but also inevitably “that participation is layered on top of existing structures and processes” [24], p. 5, we must find ways of undertaking co-production that are pragmatic within the limitations inherent in prevailing academic institutional contexts. This paper aims to make a methodological contribution to the literature and practice of co-production in research by demonstrating how co-production can be possible within current constraints. To do so, we take a recent research project with communities of older people as a case study. The project was conducted by the authors and provided a key opportunity for reflexive learning about co-production. We first describe the elements of co-production of the study and then assess the research using the set of expectations for co-production in research with older adults outlined by James and Buffel [5]. In doing so, we aim to offer insights on specific approaches and practical considerations that may be of benefit to other researchers, before discussing the theoretical implications with reference to the existing literature. There is no universally applicable recipe for co-production—its choice and form will always depend on the context and research question [21,24]. Offered here is empirical insight into approaches that can be considered for future co-research so that further enquiry continues to develop pragmatic approaches to co-production. Our research question within this paper is as follows: How can the expectations of co-production be met within the constraints of current academic systems?

We would like to emphasise at the outset that, in our opinion, the funder of the case study research has been an excellent example of a supportive, progressive funder who set high standards for the research to meet a forward-thinking strategy for social good. In the Discussion, we highlight ways in which they have actively supported co-production. The critiques we make in addition are not of this funder, but of the constraints and inequities in the current context of funding of academic research in the UK.

## 2. Materials and Methods

This paper presents a reflexive review of an empirical case study. We begin this section by describing the case study project to provide context and to highlight the elements contributing to co-production. The research objective of the case study was to develop an understanding of factors contributing to socially resilient older communities. The study was a 30-month project finishing in 2024, conducted with seven almshouse charities around England that offer affordable accommodation to older people in need (see [28] for further explanation of almshouses). We refer to these organisations as our ‘[research] partners’. Our partners had provided for those in need for an average of 250 years. We approached the research from the perspective that the partners were exemplars of resilient communities and that their members were experts by experience. Based on discussions relating to the aims and resources of each partner, three partners nominated a member of staff to act as a project liaison (partner project liaison)

Governance of the project comprised the following:A core project team of four academic researchers (two on short-term contracts) and the project liaison from one of the research partners (five members in total);A full project team of the core project team plus the remaining two project liaison colleagues (seven members in total);A Residents Advisory Group (RAG) for the project of almshouse residents from four partners (11 members in total);A Professional Advisory Group (PAG) for the project of 21 members with experts from housing, law, almshouses, and EDI (equality, diversity, and inclusion).

The core project team met at least weekly over the first eighteen months, and fortnightly thereafter. The full project team met monthly over the first eighteen months and bi-monthly thereafter. The PAG met online three times. The RAG met four times in person at a different almshouse site each time, with an average attendance of 8 out of 11 members. These meetings were guided by an independent facilitator and one academic member of the research team. We followed an ethic of care [29] to ensure that the RAG members’ preferences and capabilities were central in organising meetings (for example, arranging taxis and catering).

Beyond involvement in project governance, residents, staff, and trustees participated in data collection as interviewees. Recognising the limited rights to involvement of many almshouse residents [30], the study aimed to emphasise residents’ voices. The primary research method used was semi-structured interviews (*n* = 49) and three focus groups with almshouse residents (*n* = 16) with an age range of 54 to 97 years. This was complemented by semi-structured interviews with staff members (senior staff *n* = 16, operational staff *n* = 8) and partner trustees (*n* = 13); document analysis, including the partners’ websites, resident selection processes, and annual reports; and site visits (*n* > 25), with ad hoc conversations written up as field notes, and photos taken of the sites, facilities, and relevant materials, such as notice boards. Thus, the research project involved co-production both with organisations providing services for older people and with older individuals themselves. We use the term ‘participant’ below for those who contributed specifically to data collection and not project governance. The study was approved by the university research ethics committee of the lead author.

In order to provide early findings for discussion with stakeholders, data collection was divided into two phases. Phase 1 comprised interviews with four partners. Analysis of this material was the basis for a summary document of preliminary findings. This was sent by email and presented to the full project team, the RAG, and the PAG. Phase 2 comprised interviews, which were lightly revised based on learning from Phase 1, with the remaining three partners and the focus groups of residents.

Near the end of the Phase 2 analysis, a day-long workshop was held, to which all RAG, PAG, and project team members were invited. As a result of the workshop, the themes were restructured and then finalised before being published on an online, free-to-access Knowledge Hub (website). This was the primary output. Secondary outputs to date include three papers submitted to peer-reviewed journals, two of which include partner project liaison colleagues as co-authors.

## 3. Results

In this section, we systematically describe the co-production in the study. We draw on the approach taken by O’Mara-Eves and colleagues [14] and James and Buffel [5] to summarise the role of each non-academic co-researcher group for each stage of the research (see Table 1). We consider five stages in the research: the funding bid and research design; participant recruitment; data collection; data analysis and dissemination. We then describe each stage in more detail to explain what was undertaken and why, what benefits were found, and what limitations were experienced. We subsequently present a short narrative reflection based on the synthesis by James and Buffel [5] of co-production in research with older adults.

### 3.1. Funding Bid Including Research Design

Triggered by the funding call, the bid team was formed, consisting of three academic researchers and the Head of Research at an almshouse charity acting as a partner project liaison. The team was multi-disciplinary, with backgrounds in almshouse research, architecture, and environmental psychology. The funder made clear that they expected some form of co-production in the proposed studies. All three academic researchers had existing links with almshouse charities (in two cases, developed outside of academia), and these links enabled the team to include one charity’s Head of Research as a fully involved member of the bid team and to gain support from a further six housing charities. Discussions of the different research interests in the team coalesced around a research question proposed by the non-academic member. When the research question and general approach were developed, they were discussed with three older almshouse residents. This was undertaken in part to comply with the funder’s requirement for PPI. Insights were gained from this input which helped to refine aspects of the proposal, specifically on using new technology. In addition, the team held informal discussions with an operational member of staff at one of the partners and the CEO at another: this provided useful insights on practical factors which helped to refine the methods to be used. Building on existing relationships developed over several years, the team involved senior staff of potential partners at an early stage, primarily Chief Executive Officers or equivalent, to establish an interest in collaboration. When the near-final proposal was prepared, the CEOs provided letters of support demonstrating collaboration, which is likely to have improved the likelihood of winning funding.

From this description, it is clear that the decision-making for the research design remained with the bid team. Nevertheless, the involvement of a non-academic co-researcher on the team was critical and set the direction of the research. Although involvement put extensive time demands on the partner project liaison colleague, the collaboration worked because securing funding for charity–academia research was part of their remit as Head of Research. Because of the tight time constraints on the funding call, and because the call required a time-limited proposal presented in some detail, it was not feasible to share decision-making outside the bid team.

### 3.2. Participant Recruitment

Detailed plans for participant recruitment were discussed by the core and full project team. This enabled valuable input from the partner liaison members. The RAG gave advice on the best way to recruit participants, contributed to a recruitment approach customised for each housing site, and also publicised the project to their community within each almshouse charity. In addition, informal consultation with key operational staff, as advised by the partner project liaison members, facilitated the planning of multiple channels for the recruitment of resident participants specific to each housing site, including flyers for noticeboards, notices in online newsletters, and tea/coffee mornings at which the academic researchers could present the project and ask for participation. We were mindful of a previous critical comment on the role of ‘gatekeepers’ as having the potential to hinder, as well as facilitate, participation [15], and did not ask the partner liaison members or operational staff to approach residents directly regarding engagement in the research. Similarly, senior staff were consulted for suggestions of operational staff and trustees to approach but did not take part directly in recruitment. This contributed to the rigour of the study in ensuring that participants had volunteered freely. It also provided ‘safe spaces’ [15,31] in which participants could be critical about aspects of their experiences if they chose to, and some did.

No monetary incentive was offered to staff or trustees, as they were participating as part of their professional roles. Remuneration of a shopping voucher worth GBP 10 (EUR 12/USD 12) was offered to each resident at the end of the interview, focus group, or RAG meeting. In fact, the study budgeted for a payment of GBP 25 per hour equivalent, in line with recommendations from the national health and care research funding body [32]. However, from discussions within the project team and checking with government documentation, it became clear that any recompense paid to participants receiving state welfare benefits could jeopardise these welfare payments. Because our intended participants were likely to be in receipt of means-tested state benefits, including housing benefit, and because individual expert financial advice would be needed in every case, we realised that we could not pay the full rate. With the help of the partner liaison members on the full project team, we reached a workable compromise of a small voucher worth GBP 10 offered directly to each resident who participated, with the remainder of the recommended hourly rate being paid to the almshouse charity with the agreement that the funds would be paid into a social fund benefitting all residents. RAG members determined how the community fund would be distributed at their housing site. The arrangement applied for participation in interviews, focus groups, and the RAG.

### 3.3. Data Collection

The three partner project liaison team members facilitated data collection over the duration of the project, advising on upcoming local events or opportunities for recruitment, for example. At the first RAG meeting, members piloted the interview questions and made substantial changes to the interview guide before resident interviews began. Their contributions included identifying questions that were too vague and would have diverted the focus, suggesting alternative, more targeted questions instead; providing alternative wording for questions they felt would be confusing and shaping the language used in the interviews to be most accessible to their peers; reordering the flow of questions; and identifying additional questions. The benefits to the project included customised recruitment in line with local events and an interview protocol that flowed more smoothly, which is likely to have helped to build rapport and to result in richer data.

### 3.4. Data Analysis

Due to the project time constraints, data analysis was conducted by the academic team members. In order to involve research partners in the process, the project was structured into two phases. Data collection from four out of seven research partners was conducted during Phase 1. The data were analysed and the preliminary findings were provided to the full project team, the RAG, and the PAG for review and comment. The responses from the research partners were brief but universally supportive, indicating that the research approach and first findings were acceptable. In Phase 2, data collection with the remaining three research partners was completed and analysed. A selection of the findings was presented at a day-long workshop to which the RAG and PAG members, as well as some of the senior and operational staff and trustees from all seven research partners, were invited. There were 31 attendees, with good representation across partners and across residents. The RAG members helped to plan this workshop, including offering advice on what information to include, how to structure breakout groups, what type of lunch would be preferred by older participants, and maintaining a respectful atmosphere. Because of the scale of the project findings, it was only possible to present a subset (13 out of 30 themes) to enable detailed discussion over the course of the day. The attendees engaged fully and, as a result of the discussions, the clustering of the themes was revisited after the workshop and changed to categories that were more meaningful to the non-academic co-researchers.

The ‘power’ inherent in data analysis remained with the academic co-researchers. We were mindful of the length of time it takes to gain experience in qualitative analysis and the volume of the findings. Our solution aimed to balance input from the non-academic co-researchers without making extensive demands on their time, with a focused desire to produce project outputs and make them available in a timely way.

### 3.5. Dissemination

Despite best intentions to maintain the involvement of the partner project liaison colleagues until the end of the project, in reality, engagement came to a gradual end after about two years. There were clear organisational reasons for this, including staff turnover, promotion in seniority, significantly increased workloads, and demands on their time from other research projects and from operational needs. Nonetheless, the partner project liaison co-researchers were co-authors of academic outputs and contributed to reviewing papers. The RAG members and the workshop attendees provided guidance on the best medium and content for the primary non-academic project output, the Knowledge Hub. The partners supported the dissemination of the Knowledge Hub through their communications teams, endorsing the final output. The Knowledge Hub benefited from the guidance of RAG members and the workshop, resulting in a more appropriate structure and ensuring that language was accessible to residents as well as staff. The success of the approach was validated in meetings with residents to present the Knowledge Hub after its publication online: residents found it easy to use and clearly written.

## 4. Reflection

To assess to what extent our project met the expectations of co-research, we now apply the categories of importance identified by James and Buffel [5] and follow their terminology. These categories comprised four areas of potential benefit: (i) achieving a deeper understanding and improved data; (ii) developing the skills and experiences of co-researchers; (iii) creating change in policy, services, practice, and the community; and (iv) challenging power dynamics; and three areas of risk: (v) power differentials and risk of reproducing inequalities; (vi) different expectations; and (vii) sustainability over time. We offer brief descriptive evidence from the case study of the extent to which the benefits were or were not achieved and challenges overcome.

*Benefit 1: Achieving a deeper understanding and improved data.* By setting an initial objective of placing residents’ voices as central, and by the triangulation of data across sources (residents, operational staff, senior staff, trustees) and across sites, the resulting data set was internally consistent and extensive. This enabled the exploration of multiple perspectives on the research question of the case study, offering the potential for rich data and deep analysis. In particular, co-production enabled the project to hear feedback from participants on multiple occasions (meetings of project team, RAG, and PAG; workshop). The validation of the findings by residents and staff, at the preliminary and final stages and after publication of the primary output, supports our claim that the findings provided a deep understanding of residents’ experiences.

*Benefit 2: Developing skills and experiences of co-researchers.* This expectation encompasses new experiences, building relationships and networks, and developing new skills, including advocacy. The RAG members expressed their enjoyment in taking part in the study and their desire to be involved in future research. Evidence of the relationships which developed included inter-site visits outside of the project. Some voiced their intention to advocate for additional facilities at their housing site based on discussions with other RAG members. Resident-led initiatives, including the creation of a Residents’ Forum at one location and diversity training at another, were outcomes from the project. This evidence shows the development of skills, including an increase in self-efficacy and skills in self-advocacy [15], through co-production. Relationships developed between the partner organisations too, including introductions at the workshop and a joint working session to share knowledge on diversity policies. Some of the partner project liaison colleagues described themselves as becoming more self-confident in being involved in academic research and expressed their intention to be co-producers of future research.

*Benefit 3: Creating change in policy, services, practice, and the community.* There was evidence of potential change resulting from involvement in co-production. The session on diversity policies, which was written up as a case study as part of the Knowledge Hub, was seen by partners as providing valuable insights on the experience of others and resulted in at least one partner making additional formal commitments to diversity and another taking steps towards a diversity policy. In addition, one partner noted an increased focus on communication with residents as a result of their involvement in the project. These changes additionally illustrate enhanced skills among the research partners.

*Benefit 4: Challenging power dynamics.* One change, brought about by the project and with the potential to challenge existing power dynamics, was the application from an RAG member to be a member of the board of a key sector organisation: the first application from a resident. A second was the initiation of a Residents’ Forum at one research partner, which will ensure a greater say from residents about housing, facilities, and services. In addition, over the course of the project, we noted the first involvement of residents as speakers in events organised by a key sector organisation. This is a further indication of the ripples of empowerment which can spread from co-produced research.

*Challenge 1: Power differentials and risk of reproducing inequalities.* We cannot claim to have set aside power differentials either in the project’s context or between co-researchers on the project other than partially. However, by including residents both as our primary information sources and as part of project governance via the RAG, we minimised the risk identified by Corrado et al. [23] and others of reproducing existing inequalities within a group. Although it is likely that some RAG members may have been individuals more likely to be engaged in advocacy for their own needs outside the project, our data were drawn from across the resident communities. As discussed above, our attempt to ensure appropriate compensation to encourage participation ended up as a compromise. In contrast to partial change in the power hierarchy between academia and non-academic participants, the culturally normal power differentials and inequalities between the academic researchers were reproduced during the project. The early-career researcher conducted all of the interviews and focus groups with residents and co-facilitated the RAG. A primary reason for this allocation of responsibility was the funding structure: only the early-career researcher (on a short-term contract) was funded full-time on the project and the project funding was time-limited.

*Challenge 2: Different expectations.* The second challenge concerned the level of participation, i.e., that “active involvement was not always what older people wanted or were able to commit to” [5], p. 2949. In addition, for employees at partner organisations, research involvement was rarely their ‘day job’ and operational needs had to take priority. Here, we believe that the light-touch approach worked well. The mindful planning of each individual’s involvement meant we maintained the input from the RAG and succeeded in extensive data collection. The one area where this was limited was the long-term engagement of partner liaison colleagues, as explained above. This aligns with the criticism by Oliver and colleagues [13] that research planning needs to consider the time demands of co-production on all stakeholders.

*Challenge 3: Sustainability over time.* The third and final challenge was that of sustainability, i.e., continuing to maintain relationships and the resources required for research and dissemination. We found that, despite their willingness to collaborate, our partners could not support the project over its full duration. In addition, our project suffered the common frustrations of being time- and funding-limited and the only full-time researcher on the team being on a short-term contract. Through their work in facilitating the RAG and in conducting the interviews with residents, this researcher built strong connections with the partner communities over the course of the project. A second researcher who undertook the staff and trustee interviews was also on a short-term contract. The loss of these researchers at the end of their contracts meant that the valuable relationships with members of the partner communities were also lost, to the detriment of future research. The project-based nature of research funding also has the potential to undermine the willingness of non-academic co-researchers to become involved in further research when their involvement in a project, and the connections built in the project team, come to an abrupt and final stop. As has been the case with other co-produced studies [29], the continuing impact of the project will not be tracked now that funding has ended.

## 5. Discussion

In our case study, the non-academic enactors of co-production were the funder, the partner project liaison colleagues, the RAG and residents, the partners’ operational and senior staff and trustees, and the PAG. The contributions of each will now be discussed.

The funder was a key actor in creating the context in which the research was conducted, and research context has been established as a major determinant of the extent to which co-production can be realised [21,22]. In the case study, the funder set a clear expectation that proposals would demonstrate close collaboration, including by requiring PPI on the proposal itself. The bid team was in a position to respond to these requirements due to the relationships they had already developed with almshouse charities, in some cases over years and outside of academia. This was further significantly facilitated by a partner appointing their Head of Research to the team in a project liaison role, with objectives aligned with those of the academic team members to deliver co-produced research. Thus, the context [21,24], in terms of the alignment of the funder’s requirements and the resourcing and objectives of a non-academic partner, enabled successful co-production.

The aims of the non-academic partners and their practical commitment to the research were key. The involvement of the partners’ project liaison colleagues as team members kept the needs, aims, and capabilities of the partners at the forefront of the project throughout. It also facilitated reciprocal learning, usually via the regular full project team meetings. This was possible because the research partners allocated responsibility and committed resources to the project. The guidance from the project liaison team members had practical value, for example, in helping to resolve the issue of compensation for participants. At the bid stage, the input of a partner liaison colleague was vital in enabling a research focus which addressed the interests of the partners, as well as those of the academic researchers. However, involvement in academic research is resource-intensive and protracted over time. Despite their willingness to collaborate, the case study demonstrated that the demands of research are such that many organisations, particularly charities and smaller groups, simply cannot support projects in an extensive way for an extended period of time. For successful co-production, the operational and other demands on co-researchers, at an organisational and an individual level, must be considered, as Oliver and colleagues [13] also argue.

The RAG benefited the project through bolstering recruitment, piloting the interview protocol, suggesting improvements that led to richer data, validating the findings, and advising on dissemination. The project benefited the RAG members through their enjoyment of engagement, opportunities for travel to other almshouse sites, meeting new people and forging new relationships, and developing their voices for self-advocacy. The small numbers of meetings of the RAG succeeded in ensuring sustained involvement for the duration of the project and did not ask more engagement than the members were able to commit to [5]. The independent facilitation of RAG meetings succeeded in providing a safe space for open discussion. We propose that an RAG or equivalent may be a light-touch and feasible element to include in research with older adults to bring benefits to non-academic groups, as well as enhance the research outputs.

Both senior staff and operational staff played enabling roles: senior staff by supporting the bid and outputs, and operational staff by offering on-the-ground advice. Both levels of staff opened up channels for recruitment, contributed to the validation of the findings, and advised on dissemination. Crucially, staff participation in interviews provided triangulation for the findings so that the project outputs could resonate with residents and those who support their needs. The PAG extended the project with perspectives from the wider sectors of housing and communities of older people, as well as legal and diversity concerns, providing a more balanced perspective on the project and its outputs. Co-production places demands on time and involvement which may be beyond what is feasible for individuals and senior and operational staff of smaller organisations to support [13]. Again, light-touch involvement can be proposed as a way of incorporating collaboration. While it may not lead to full and equal decision-making, it ensures at least some input and guidance from experts with experience outside of academia.

One objective of the case study project had been to place the voice of residents centrally. The difficulties surrounding payment of recompense was a challenge, and official guidance placed the burden on the participant to investigate the impact of accepting recompense for their role in research [32]. While a compromise was reached in the case study so that residents would benefit from the research budget in a manner commensurate with their involvement, the societal powerlessness of our participants was laid bare. There may be analogous issues with marginalised groups in other research, also requiring bespoke solutions. The current rules on recompense in the UK serve to systematically exclude the voices of the less affluent, and specifically of less-affluent older people, from participation in research. Co-production is inevitably situated within the power structures of society, and there is a need for greater acknowledgement in research of how the government, academia, and other social institutions constrain who can participate in research [25]. Although some might consider that rules on state benefits are not an academic constraint, we argue that such rules may limit who can act as co-producers of research outputs and therefore what academic research is possible.

The practical ways in which co-production occurred in the case study project were shaped by other constraints inherent in current academic orthodoxies relating to research practice and careers. One constraint is the project-based approach to research, which Oliver and colleagues [13], p. 4, call a “culture of hit-and-run research”. It demands early specification of outputs and places responsibility for their delivery on the Principal Investigator. In this context, decision-making by co-researchers at the bid stage must be limited to the individuals responsible for delivery [33], and this necessarily extends through the course of the research. In the case study, the bid was developed within the core project team only, and data analysis was conducted by the academic researchers only for these reasons. The project-based approach makes little allowance for training or for the competing demands on non-academic co-researchers in organisational positions in the context of their lives [29]. On this project of 30 months, with one full-time and three part-time academic researchers, the breadth and volume of outputs meant that we could not expect our co-researchers to spend the days or even weeks needed to review the content in detail.

While the project succeeded to a limited extent in sharing power with the members of almshouse communities, especially the older residents, it did not challenge the academic power hierarchy. In considering the imbalances of power in co-production, we need to consider the marginalised within academia, especially early-career researchers who are disempowered within career and funding structures. Boylan and colleagues [26] argue that those actually ‘doing co-production’ are often at the bottom of the academic hierarchy, and this was true on our project. The precarity of employment in working on time-bound projects within a short-term contract meant that the junior researcher remained disempowered at the end of the project, being forced to seek another contract without the opportunity to consolidate learning, to develop capacity in a specific domain, or to maintain the relationships developed with the research partners and participants. The contribution of early-career researchers is stymied by their powerless position as contract labour.

The nature of academic funding as project-based and time-limited constrains the potential for the development and maintenance of relationships with co-researchers outside of academia [1,25], and this was true for the case study. There is frequently an unarticulated expectation from funders that academic team members will continue to work on outputs and to cultivate relationships beyond the project but without payment or recognition [26]. This is a further way in which power in the form of resources is used in a way that privileges some and disempowers others. Only those team members with a long-term employment contract and a workload that allows sufficient time can continue to work on papers for publication (the currency of an academic career), or present findings at conferences, without funding. At the end of a project, those on short-term contracts tend to be either job hunting or beginning again on another project without time to consolidate the expertise from the original project. The valuable relationships created during the project cannot be sustained and knowledge and opportunities for learning are lost, significantly reducing the longer-term benefits of research.

It was noted above that the bid addressed a research question posed by the non-academic co-researcher on the bid team, but which aligned with the research interests of the academic members. This is a key constraint for academic researchers. As argued by Pearce [25], power differentials cannot be set aside for co-production work while required in all other aspects of career performance. Academic careers require a coherent body of research demonstrating the development of expertise in a (typically very specific) domain: this is part of the performative logic discussed by Paylor and McKevitt [1]. In order to publish in peer-reviewed journals, an academic researcher is required to ground their study on existing published papers, with a requirement from many editors to reference current discourse. This means that academic researchers are usually not free to pursue any arbitrary question, but must find a way to bring together current discourse in the academic literature with the needs of non-academic partners. The approach we found in this study offered a space (albeit limited) for genuine power sharing in a team in which the research question was ultimately led by a non-academic member but guided by discussions in which academic researchers could identify current narratives, concepts, and gaps in the literature.

In this reflective case study review, there are limitations in the approach we have taken. Other analyses of the aims of co-production (e.g., [14,15]) could be used for reflection. As we noted in the Introduction, the choices and forms of co-production will always depend on the context and research question. The settings of our seven almshouse charity partners may not generalise to other communities of older people, limiting the transferability of some of the insights offered. A key limitation is that this paper presents a self-assessment. As such, it is necessarily open to a subjective slant, but it does offer, we hope, a valuable reflection on our research. We would like to see future projects offer their own reflections to provide the bases for review, as outlined in the work by James and Buffel [5], to help to develop the domain of co-produced research with older adults.

## 6. Conclusions

In order to answer the question this paper posed—‘How can the expectations of co-production be met within the constraints of current academic systems?’—using research that we had conducted on social resilience in communities of older adults as a case study, we reflected on lessons for co-production. Combining the lived expertise of the members of almshouse communities with insights from the academic literature, the project drew on co-researchers’ knowledge to understand the factors contributing to community resilience. In this way, the project preserved community knowledge, made explicit tacit ways of knowing and doing, and documented these for the wider community. As such, it can be considered a successful example of co-production at a general level. The elements of co-production at each stage of the research process were then examined, and then assessed against the expectations mapped by James and Buffel [5] in their systematic review of co-production in research with older adults. We propose that the overarching aim of co-production—of sharing power and challenging power structures [7,8]—was met, albeit in a limited way, while the main expectations were met by the approach, research design, and governance of the project. We conclude that the study incorporated rich co-production with co-authoring according to the categorisation of the Co-production Collective [14]. While the idealist aims for co-production should continue to drive research towards deeper collaboration beyond academia, we also need pragmatic ways to build towards the ideal within the constraints of today’s academic systems: the current study demonstrates some of these practical mechanisms.

Although we acknowledge that the case study did not achieve equality between co-researchers for the contextual reasons discussed above, the ideals of co-production should continue to inspire us to look for practical ways in which power can be shared and marginalised voices can be heard. Based on key points from our discussion above, we propose four recommendations relating to academic constraints from a perspective of critical utopianism, which “knows that reality can be altered for the better but understands the difficulties in doing so” [3], p. 108.

Academia in the UK must make representations to change the rules which prevent people in receipt of means-tested state benefits from recompense for their involvement in research. Future national policy needs input from research with all citizens, especially the disempowered.We would like to see the rapid phasing out of short-term research contracts so that the precarity of employment no longer undermines the capacity of early-career researchers to develop to their full potential. This would also significantly improve the issue of the sustainability of co-production described above.A more fundamental shift is needed to move away from the inappropriate expectation that knowledge generation and learning can be achieved through time-limited projects. A first step would be for funders to continue to fund project members, at least partially, for a minimum of 12 months beyond the end of a specific deliverable—what Staniszewska and colleagues call ‘glue’ money [7]. Academic culture is such that, if they are in academic employment and if their institution grants them the time, project members will continue to work on the project (revising and resubmitting papers; maintaining the relationships outside of academia; undertaking dissemination; facilitating impact), so this work should be funded and acknowledged.Finally, there is a need to transform academic culture by establishing meaningful acknowledgement for the development of long-term relationships with groups and communities outside of academia, at least in social science departments. Genuine, rich, and deep co-production can only be achieved by developing the expectation that, to undertake academic research with people, researchers need to be engaged and equal partners in communities outside of academia.

Our four recommendations here are aspirational. In the meantime, we encourage future research to seek ways in which aspects of co-production can be introduced despite the constraints of current academic systems, to the benefit of research, co-researchers, and society.

## Figures and Tables

**Table 1 ijerph-21-01503-t001:** Summary of stakeholder involvement in the research process.

	Partner Project Liaison Colleagues	Residents (Including RAG Members)	Operational Staff	Senior Staff and Trustees	Sector Professionals in PAG
**Funding bid including research design**	One colleague was a member of the bid team	PPI review of early draft	Informal discussion with one partner	Informal discussion with one partner; letters of support	Letters of support
**Participant recruitment**	Three colleagues attended team meetings which planned recruitment	RAG assisted in their own almshouse community	Advised and facilitated	Limited to senior staff suggesting trustees and operational staff to approach	Not involved
**Data collection**	Three colleagues facilitated	RAG reviewed and piloted interview and focus group protocols. Interviews/focus groups with *n* = 65	Interviews with *n* = 8	Interviews with *n* = 16	Not involved
**Data analysis**	Reviewed preliminary findings and took part in workshop to review final findings before dissemination	RAG reviewed preliminary findings and took part in workshop to review final findings before dissemination	Took part in workshop to review final findings before dissemination	Took part in workshop to review final findings before dissemination	Took part in workshop to review final findings before dissemination
**Dissemination**	Co-authors on academic papers	Discussed the medium and content for outputs; disseminated to residents within their own and nearby almshouses	Communications team publicised the final output	Some provided endorsement of the final output	Some provided endorsement of the final output

Notes: PPI—patient and public involvement—a process by which (usually healthcare) users contribute to research; RAG—Residents Advisory Group for the project; PAG—Professional Advisory Group for the project.

## Data Availability

The paper is a methodological reflection and, as such, does not draw on primary empirical data.

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
