# Peer review of "Co-Production Within Academic Constraints: Insights from a Case Study"

_ijerph, 2024, doi:10.3390/ijerph21111503_

Round 1
Reviewer 1 Report
Comments and Suggestions for Authors
The paper explores how the expectations of co-production can be met within the constraints of current academic systems. It is well written and offers a detailed case study of using co-production throughout a research project. The authors evaluate the success of the co-production methods and conclude by discussing in more detail some of the constraints relating to the aspiration of using co-production in academic projects. The paper is potentially an important contribution to the field, it provides an honest and realistic discussion of the challenges of using co-production in academia.
I suggest a few minor ways in which the paper could be improved.
Introduction: Before you discuss the “common aspirational benefits” of co-production, it would be useful to give the definition of co-production in research that you are following. It would be useful to readers new to co-production to have a concise definition of the term (e.g. co-production is working with as opposed to for or on research participants).
Following, I suggest that the paragraph starting “Definitions of co-production vary”, (rather than definitions) talks about approaches, categories and benefits of co-production. The opening sentence should be changed to reflect this.
Methods: Perhaps Table 1 could more closely reflect the different project stakeholders outlined at the start of the methods section. For example, it would be nice to see the distinction between ‘key partners’ and ‘partner organisations’ reflected in table 1.
Results: The first part of the results is a very valuable narrative of the use of co-production throughout the different stages of the research project. The paper demonstrates some of the practical mechanisms that could be used to deliver co-production within the constraints of academic systems.
The second part, the evaluation using the systematic review from James and Buffel, could maybe do with a bit further development. As the review themes were never proposed as a mechanism for evaluation, maybe this should be framed in another way?
In relation to the benefits, can the observations reported be directly linked to the co-production project? For example: Deeper understanding of data: I’m not sure how this can be justified if you are not comparing it to previous research or a baseline review of some sort – deeper than what?
When you talked about developing skills of co-researchers you only mention the older residents. In the methods you stated that “the research project involved co-production both with organisations providing services for older people and with older individuals themselves.” It would be good to hear how the organisations’ skills were developed.
In general (in part I imagine due to word count limitations) the evaluation section is very brief and claims are made without sufficiently demonstrating the evidence behind them. Perhaps this section should just focus on the challenges of co-production identified by James and Buffel, giving more space for discussion of these points.
Alternatively, if you want to keep in the evaluation of how the case study demonstrated the benefits of co-production, perhaps edit the methods/results to reduce some of the repetition between these two sections (e.g. Remuneration of a shopping voucher worth £10 (€12/$12) 300 was offered to each resident at the end of the interview – mentioned in both; details of the second stage workshop – mentioned in both).
Discussion: The discussion brings the paper deep into the considerations that are needed when conducting co-production in academia. The paper clearly sets out some of the constraints of the current UK system linked to time-limited funding, short term contracts, and engagement with non-academic communities. These points are well made, but perhaps there should be a little bit more of background context in the introduction, and the implications in the methods and results. At the moment these points feel a little be disjointed from the rest of the paper.
Author Response
Reviewer #1
Thank you very much for your thoughtful review. We found your comments very helpful in improving the paper. We have addressed each one as we detail below (our responses in italics, extracts from the revised MS in red).
Introduction: Before you discuss the “common aspirational benefits” of co-production, it would be useful to give the definition of co-production in research that you are following. It would be useful to readers new to co-production to have a concise definition of the term (e.g. co-production is working with as opposed to for or on research participants).
Thank you – we agree that an early definition is appropriate. We’ve added In essence, co-production is a participatory approach to research (Paylor & McKevitt, 2019) which aims to work with (as opposed to on or for) non-academic participants. Co-production approaches aim to render research more democratic after the first sentence (ll. 33-35).
Following, I suggest that the paragraph starting “Definitions of co-production vary”, (rather than definitions) talks about approaches, categories and benefits of co-production. The opening sentence should be changed to reflect this.
Agreed. We have replaced ‘Definitions of co-production vary’ with Approaches to co-production vary in terms of meaning, scope and aims. (l.108).
Methods: Perhaps Table 1 could more closely reflect the different project stakeholders outlined at the start of the methods section. For example, it would be nice to see the distinction between ‘key partners’ and ‘partner organisations’ reflected in table 1.
Thank you for pointing out that we had used different terms in different places which was confusing. We have now simplified by removing references to ‘key partners’ and clarified by adding We refer to these organisations as our ‘[research] partners’ (l 226). We have made all references to Partner Project Liaison consistent throughout, and ensured that that we use ‘partner’ consistently for the organisation and ‘colleague’ consistently when we are referring to a person. We have updated the column headings in Table 1 for greater clarity (l. 309).
Results: The first part of the results is a very valuable narrative of the use of co-production throughout the different stages of the research project. The paper demonstrates some of the practical mechanisms that could be used to deliver co-production within the constraints of academic systems.
The second part, the evaluation using the systematic review from James and Buffel, could maybe do with a bit further development. As the review themes were never proposed as a mechanism for evaluation, maybe this should be framed in another way?
Thank you for this insightful comment. As a result, we have shifted the focus of the paper to become a reflection on the case study instead of an evaluation. We now clarify that we use James and Buffel for guidance rather than as a evaluation framework. We have changed the paper title to replace ‘evaluation’ with ‘insights’, and replaced ‘evaluation’ with ‘reflection’ throughout as appropriate. In the Results section, we use the subheading ‘Reflection’ in place of ‘Evaluation’ (l. 468), replace ‘evaluate’ with ‘assess’ (l. 469) and position the reflection as descriptive: We offer brief descriptive evidence from the case study of the extent to which benefits were or were not achieved and challenges overcome (ll. 475-477).
In relation to the benefits, can the observations reported be directly linked to the co-production project? For example: Deeper understanding of data: I’m not sure how this can be justified if you are not comparing it to previous research or a baseline review of some sort – deeper than what?
We agree with your comment – it’s not a comparative study. As noted above, we now position the reflection as descriptive. The phrase ‘Deeper understanding of data’ at l. 471 and 478 is taken directly from James and Buffel so we clarify this at ll. 469-471: we now apply the categories of importance identified by James and Buffel (2023), and follow their terminology. We have removed another case of inappropriate use of ‘deeper understanding’ in the Abstract. We have also checked to ensure that we are using appropriate and not deterministic or comparative language in the other sections on benefits.
When you talked about developing skills of co-researchers you only mention the older residents. In the methods you stated that “the research project involved co-production both with organisations providing services for older people and with older individuals themselves.” It would be good to hear how the organisations’ skills were developed.
We have added the following to Benefit 2: Some of the partner project liaison colleagues described themselves as becoming more self-confident in being involved in academic research and expressed their intention to be co-producers of future research. (ll. 503-505). Under Benefit 3, we have clarified that changes to commitments to diversity, diversity policy and communication are also examples of development of organisational skills: These changes additionally illustrate enhanced skills among the research partners. (l. 512-513).
In general (in part I imagine due to word count limitations) the evaluation section is very brief and claims are made without sufficiently demonstrating the evidence behind them. Perhaps this section should just focus on the challenges of co-production identified by James and Buffel, giving more space for discussion of these points.
Thank you for this suggestion. As set out above, we have shifted the focus of the paper from ‘evaluation’ to ‘reflection’. In addition to positioning the text here as descriptive (ll. 475-477), we have extended the discussion at several points (ll. 481-488, 503-505, 510, 512-513, 530-537, 552-554, 561-570).
Alternatively, if you want to keep in the evaluation of how the case study demonstrated the benefits of co-production, perhaps edit the methods/results to reduce some of the repetition between these two sections (e.g. Remuneration of a shopping voucher worth £10 (€12/$12) 300 was offered to each resident at the end of the interview – mentioned in both; details of the second stage workshop – mentioned in both).
Thank you for pointing out the inadvertent repetition – this has been removed.
Discussion: The discussion brings the paper deep into the considerations that are needed when conducting co-production in academia. The paper clearly sets out some of the constraints of the current UK system linked to time-limited funding, short term contracts, and engagement with non-academic communities. These points are well made, but perhaps there should be a little bit more of background context in the introduction, and the implications in the methods and results. At the moment these points feel a little be disjointed from the rest of the paper.
A useful and insightful comment – thank you. We have added in more detail on academic constraints in the Introduction (ll. 165-175) and in the Results section (ll. 424, 530-536, 552-554, 561-570).
Reviewer 2 Report
Comments and Suggestions for Authors
The paper explores issues relating to the use of co-production in academic research. The article looks at these in the context of 'potential constraints' within academic settings. This was done a via case study examining the issue of social resilience in English almshouse communities. The co-production approach has become increasingly popular in academic work, notably in the field of health and social care. This paper makes an interesting contribution to the literature but there are some concerns which can also be identified:
1. The paper needs a clear introduction setting out the aims and objectives of the paper - essentially developing the first paragraph of page 2 and then having a new subhead to examine the potential benefits of co-production.
2. Another sub-head was also needed on page 3 where the discussion shifts to the problems of co-production, with perhaps more stress on the potential issues of power imbalances between the different groups involved in co-production. Generally, I think pages 2-3 could do with some editing to sharpen the points which are being made. There also needs to be some re-organisation of this opening section as the first paragraph of page 4 seems to relate more to methodology and should be placed under a new section.
4. The opening paragraph on 'Materials and Methods' needed a much clearer statement of the research questions under investigation. The research objective of understanding 'factors contributing to socially-resilient older communities' was confusing as the paper seemed more about the merits and de-merits of co-production (as explained on page 8) or about the 'constraints of current academic systems'.
5. Second para of page 5 sets out details of the samples used for the study but a table here would have been helpful.
6. Page 8 notes that the data analysis was conducted by the research team. This needs to be explained as a co-production approach would typically involve all relevant partners.
7. The benefits of co-production outline on pages 8-9 are interesting but could have done with some illustrative quotes to support some of the findings.
8. Page 10. A research question is introduced under the discussion but this needed to come in much earlier and it still left a sense of confusion about the focus of the paper.
9. Page 12 outlines concerns about the position of contract researchers which I agree with but fit awkwardly in the context of the paper, as do the recommendations. On recommendation 3, many grant bodies are flexible about funding; the cost implications of compulsory 12-month extensions, would be challenging.

Author Response
Reviewer #2
Thank you very much for your review which has helped us to improve the paper. We provide a detailed response to each point below (our responses in italics, extracts from the revised MS in red).
- The paper needs a clear introduction setting out the aims and objectives of the paper - essentially developing the first paragraph of page 2 and then having a new subhead to examine the potential benefits of co-production.
Thank you for this feedback. We have expanded the description of aims and objectives in Paragraph 1, adding: While there has been much theoretical discussion of co-production as we will discuss below, there is relatively little literature focusing on the pragmatic issues confronting scholars who are faced with adopting this approach to research. The current paper aims to address this gap and to present practical insights on the use of co-production in research. In particular, academic constraints to effective co-production are considered. (ll. 68-73). We have added a subheading for Benefits of co-production as suggested (l. 76) and for Aims and objectives of this paper (l. 194).
- Another sub-head was also needed on page 3 where the discussion shifts to the problems of co-production, with perhaps more stress on the potential issues of power imbalances between the different groups involved in co-production. Generally, I think pages 2-3 could do with some editing to sharpen the points which are being made. There also needs to be some re-organisation of this opening section as the first paragraph of page 4 seems to relate more to methodology and should be placed under a new section.
A useful suggestion, thank you. We have added subheadings for Meanings of co-production, Critiques of co-production and Aims and objectives of this paper (ll. 108, 136, 194). The discussion of power imbalances has been extended, with a focus on academia in keeping with the focus of the paper (ll. 165-175). We have removed some text to help to sharpen the points in this section. With the improved flow and clarity of objectives, we have retained a brief outline of the methodological approach in the Introduction section to provide the reader with an overview before moving into the detail in the Materials and Methods section.
- The opening paragraph on 'Materials and Methods' needed a much clearer statement of the research questions under investigation. The research objective of understanding 'factors contributing to socially-resilient older communities' was confusing as the paper seemed more about the merits and de-merits of co-production (as explained on page 8) or about the 'constraints of current academic systems'.
We now begin the Materials and Methods section with a statement of the paper’s focus: This paper presents a reflexive review of an empirical case study. We begin this section by describing the case study project to provide context and to highlight the elements contributing to co-production (ll. 220-222).. The research question addressed by the paper is also stated seven lines earlier in the Introduction: Our research question within this paper is: How can the expectations of co-production be met within the constraints of current academic systems? (ll. 211-212).
- Second para of page 5 sets out details of the samples used for the study but a table here would have been helpful.
Thank you for this suggestion. We have considered this suggestion but concluded that the table would simply list the sources of data so we have retained the short description in the text.
- Page 8 notes that the data analysis was conducted by the research team. This needs to be explained as a co-production approach would typically involve all relevant partners.
We have added explanation at the start of this subsection: Due to the project time constraints (l.424), in addition to the more detailed explanation at lines 449-453.
- The benefits of co-production outline on pages 8-9 are interesting but could have done with some illustrative quotes to support some of the findings.
We are unclear on this suggestion – the paper is a reflective review on the approach to the case study and so there is no systematically collected dataset on the methodology from which to use illustrative quotes. However, we have extended the sections on benefits to provide more specific examples (ll. 478-521).
- Page 10. A research question is introduced under the discussion but this needed to come in much earlier and it still left a sense of confusion about the focus of the paper.
Thank you for pointing this out. This reiterates the research question presented in the Introduction (ll. 211-212). We hope that the improvements based on your earlier suggestions (clarifying the aims and objectives early in the Introduction (ll. 68-75) and the addition of the subheading Aims and objectives of this paper (l. 194)) will make the focus of the paper clearer to the reader.
- Page 12 outlines concerns about the position of contract researchers which I agree with but fit awkwardly in the context of the paper, as do the recommendations. On recommendation 3, many grant bodies are flexible about funding; the cost implications of compulsory 12-month extensions, would be challenging.
We agree with these comments. We have revised the Abstract (l. 17), Introduction (ll. 165-175) and Results (ll. 530-536) to introduce discussion of the position of contract researchers earlier. We agree that Recommendation 3 would be challenging but it is analogous to whole life costing in construction and could become part of a transformed funding process.
Reviewer 3 Report
Comments and Suggestions for Authors
This paper explores co-production in research with and about older people. This is a very welcome paper, exploring the potential of co-production work and how it can lead to better research, data and positive impacts for the people involved. It was also critically engaged on the limitations, especially the power sharing elements of genuine co-production. It was excellent to see almshouse communities used as a case study here, as an interesting model supporting older people.
I thought there was a bit more insight to be gained in exploring the context and community-based insights from the case study. Other examples of co-production with older people – such as Robertson et al (2022) ‘‘It gives you a reason to be in this world’: the interdependency of communities, environments and social justice for quality of life in older people’ - highlight co-production work with older people and older people’s involvement in their communities has a positive impact on quality of life. I think more engagement with the case study setting itself, the context of which the co-production took place and was embedded in, and critical engagement with what that means in terms of transferability for the insights gained is needed for the paper would be helpful.
I must admit I got mixed up with the levels of co-production and who was involved in what through the paper descriptions of activities which could be more consistent. Table 1 was very helpful, but I wondered if there was a more systematic way of trying to capture that dynamic of who did what/what stage throughout the paper.
It was nice to see engagement and acknowledgement and excellent partnership working with the case study hosts, funder and academics. I wondered if this was also a positive element of the story in terms of successful co-production? In the discussion you were starting to say that it had – but more information and insight would be needed for this to be clear.
I think in the evaluation section of the paper, a few more examples would help illuminate the findings. For example, benefit 1 ‘this expectation was met’ I think needs expanding. Other sections in here have examples that bring the point to life (e.g. influence and change on diversity policies) and I think more of these examples in that part of the paper would help show impact.
There was a bit of repetition around remuneration for residents. However, I really appreciated the detail given under participant recruitment about the challenges of recompense paid to participants receiving state welfare benefits and how that could jeopardise welfare payments. Many projects have had to start from scratch learning these elements – and that a personalised and individual led process is needed within research to overcome these challenges. This is such an important practical point I would have liked to see more on this as helpful for other academics to know. Great to have this pointed out in the recommendations.
Just as a small aside, the abstract could be checked through and polished, some sentences come over a bit unclear.
Overall, I really enjoyed this paper and with some minor revisions it will be an excellent addition to our understanding and application of co-production research. Well done to the authors on a really interesting, nuanced and exciting project. I look forward to citing this paper!
Author Response
Reviewer #3
Thank you for your encouraging review, which has given us a number of suggestions for improving the paper. We have attempted to address each point as detailed below (our responses in italics, extracts from the revised MS in red).
I thought there was a bit more insight to be gained in exploring the context and community-based insights from the case study. Other examples of co-production with older people – such as Robertson et al (2022) ‘‘It gives you a reason to be in this world’: the interdependency of communities, environments and social justice for quality of life in older people’ - highlight co-production work with older people and older people’s involvement in their communities has a positive impact on quality of life. I think more engagement with the case study setting itself, the context of which the co-production took place and was embedded in, and critical engagement with what that means in terms of transferability for the insights gained is needed for the paper would be helpful.
Thank you for the great reference and your thought-provoking suggestions. It is always a challenge to balance presentation of a case study with analysis or reflection, as in this case. Here we wanted the focus to be on the methodology and its intersection with academic culture so we chose to avoid too much detail on the case study lest it distract from the paper’s focus. Based on your suggestion, we added the following to the limitations: As we noted in the Introduction, the choices and forms of co-production will always depend on the context and research question. The settings of our seven almshouse charity partners may not generalise to other communities of older people, limiting the transferability of some of the insights offered (ll. 785-788).
I must admit I got mixed up with the levels of co-production and who was involved in what through the paper descriptions of activities which could be more consistent. Table 1 was very helpful, but I wondered if there was a more systematic way of trying to capture that dynamic of who did what/what stage throughout the paper.
Thank you for this feedback. We had used different terms in different places which was confusing. We have now simplified by removing references to ‘key partners’ and clarified by adding We refer to these organisations as our ‘[research] partners’ (l 226-227). We have made all references to Partner Project Liaison consistent throughout, and ensured that that we use ‘partner’ consistently for the organisation and ‘colleague’ consistently when we are referring to a person. We have updated the column headings in Table 1 for greater clarity (l. 309). We have also added clarification before Table 1: We consider five stages in the research: the funding bid and research design; participant recruitment; data collection; data analysis and dissemination (ll. 303-304), and then explain that we will explain each stage in detail (l. 305).
It was nice to see engagement and acknowledgement and excellent partnership working with the case study hosts, funder and academics. I wondered if this was also a positive element of the story in terms of successful co-production? In the discussion you were starting to say that it had – but more information and insight would be needed for this to be clear.
An interesting and useful insight – thank you. We have revised the text to acknowledge this positive element more clearly, and to provide more information on what it meant in practice:
The context (refs) in terms of the alignment of the funder’s requirements, and the resourcing and objectives of a non-academic partner, enabled successful co-production.
The aims of non-academic partners and their practical commitment to the research was key. The involvement of the partners’ project liaison colleagues as team members kept the needs, aims and capabilities of the partners to the forefront of the project throughout. It also facilitated reciprocal learning, usually via the regular full project team meetings (ll. 585-590).
I think in the evaluation section of the paper, a few more examples would help illuminate the findings. For example, benefit 1 ‘this expectation was met’ I think needs expanding. Other sections in here have examples that bring the point to life (e.g. influence and change on diversity policies) and I think more of these examples in that part of the paper would help show impact.
Thank you for this suggestion – we have revised this section to ensure there are examples throughout (ll. 478-488, 503-505, 510-513).
There was a bit of repetition around remuneration for residents. However, I really appreciated the detail given under participant recruitment about the challenges of recompense paid to participants receiving state welfare benefits and how that could jeopardise welfare payments. Many projects have had to start from scratch learning these elements – and that a personalised and individual led process is needed within research to overcome these challenges. This is such an important practical point I would have liked to see more on this as helpful for other academics to know. Great to have this pointed out in the recommendations.
Thank you. We have removed the repetition. In the Discussion, we have revised to add more emphasis, including your insight that these kind of challenges may be faced on other projects and require a personalised process to resolve (ll. 713-715, 717-710).
Just as a small aside, the abstract could be checked through and polished, some sentences come over a bit unclear.
We have checked and sharpened the Abstract, and tried to improve the clarity.
Overall, I really enjoyed this paper and with some minor revisions it will be an excellent addition to our understanding and application of co-production research. Well done to the authors on a really interesting, nuanced and exciting project. I look forward to citing this paper!
Thank you very much!